# Immunomodulatory Properties of Masticadienonic Acid and 3α-Hydroxy Masticadienoic Acid in Dendritic Cells

**DOI:** 10.3390/molecules27041451

**Published:** 2022-02-21

**Authors:** Gabriela Piñón-Zárate, Fernanda Reyes-Riquelme, Ma Beatriz Sánchez-Monroy, Mónica Velasco-Torrez, Mariano Martínez-Vázquez, Christian Adrian Cárdenas-Monroy, Beatriz Hernandez-Téllez, Katia Jarquín-Yáñez, Miguel Ángel Herrera-Enríquez, Andrés Eliú Castell-Rodríguez

**Affiliations:** 1Facultad de Medicina, Universidad Nacional Autónoma de México, Mexico City 04510, Mexico; gabrielapinon@unam.mx (G.P.-Z.); fercha_lion@hotmail.com (F.R.-R.); mony_vt93@hotmail.com (M.V.-T.); ccardenas@unam.mx (Ch.A.C.-M.); bhernandezt@hotmail.com (B.H.-T.); katys12@hotmail.com (K.J.-Y.); mikeh@unam.mx (M.Á.H.-E.); 2Departamento de Ecología y Recursos Naturales, Facultad de Ciencias, Universidad Nacional Autónoma de México, Mexico City 04510, Mexico; beatriz.sanchez@ciencias.unam.mx; 3Departamento de Productos Naturales, Instituto de Química, Universidad Nacional Autónoma de México, Mexico City 04510, Mexico; marvaz@unam.mx

**Keywords:** dendritic cells, triterpenes, cancer

## Abstract

Dendritic cells are antigen-presenting cells, which identify and process pathogens to subsequently activate specific T lymphocytes. To regulate the immune responses, DCs have to mature by the recognition of TLR ligands, TNFα or IFNγ. These ligands have been used as adjuvants to activate DCs in situ or in vitro, with toxic effects. It has been shown that some molecules affect the immune system, e.g., Masticadienonic acid (MDA) and 3α-hydroxy masticadienoic acid (3α-OH MDA) triterpenes naturally occurring in several medicinal plants, since they activate the nitric oxide synthase in macrophages and induce T lymphocyte proliferation. The DCs maturation induced by MDA or 3a-OH MDA was determined by incubating these cells with MDA or 3α-OH MDA, and their phenotype was afterwards analyzed. The results showed that only 3α-OH MDA was able to induce DCs maturation. When mice with melanoma were inoculated with DCs/3α-OH MDA, a decreased tumor growth rate was observed along with an extended cell death area within tumors compared to mice treated with DCs incubated with MDA. In conclusion, it is proposed that 3α-OH MDA may be an immunostimulant molecule. Conversely, it is proposed that MDA may be a molecule with anti-inflammatory properties.

## 1. Introduction

Dendritic cells (DCs) may be found in every animal tissue, where they recognize any antigen in a receptor-mediated interaction. Such receptors may be the Toll-like receptors (TLR), the C-type lectins, scavenger receptors, among others. After recognizing their antigens within an inflammatory environment, DCs undergo maturation, and the expression of costimulatory (e.g., CD80, CD86) and inhibitory (e.g., CD273 and CD274) molecules are enhanced [1,2]. Subsequently, DCs migrate towards secondary lymphoid organs, activating antigen-specific T cells [3].

Adjuvants are used in immunotherapy against cancer to activate the immune system in situ. The aim is to activate DCs or, in some cases, DCs cells are activated in vitro and subsequently inoculated [4,5]. In both scenarios, DCs participate by establishing the required antitumor Th1 response to eliminate tumor cells. Immunomodulatory molecules are necessary to activate DCs adequately [2], and tumor antigens, proinflammatory cytokines and TLRs ligands have been accordingly used. However, the type of immune response depends on the molecule used [6]. For instance, aluminum salts, IL-4, and prostaglandins induce a pro-tumorigenic Th2 response [1], but not an adequate Th1 response. Thus, it is crucial to identify and study new molecules able to mediate the activation of DCs.

Triterpenes are secondary metabolites consisting of C5 isoprene units (Figure 1), and they naturally occur in some medicinal plants, representing their largest reservoir of phytochemical compounds [7]. Several triterpenes have been isolated from traditional medicinal plants, such as *Amphipterygium adstringens* Schiede Schlecht (common name: Cuachalalate), a medicinal plant of ethnobotanical importance traditionally used to treat several diseases, including gastric cancer and anti-inflammatory conditions [8]. Previous studies have shown that the major components of the *A. adstringens* cortex contain tetracyclic triterpenes such as masticadienonic acid (MDA) and 3α-hydroxy masticadienoic acid (3α-OH MDA) [9]. They induced nitric oxide production in macrophages [10] and T lymphocyte proliferation in vivo [11]. As both compounds display immunomodulatory properties, it is probable that 3α-OH MDA and MDA mediate the activation of DCs. Therefore, this work aimed to study the effect of 3α-OH MDA and MDA on DCs used within the context of an experimental immunotherapy model.

## 2. Results

### 2.1. DCs Migratory Capacity after Treatment with MDA or 3α-OH MDA

After the DCs bind to an antigen, their maturation process and their subsequent migration to the lymph nodes to activate T lymphocytes were evaluated. To study the effect of triterpenes on the DCs migration potential, we used an in vivo migration model in which a test molecule is topically applied on the mice ear skin to assess the migration of a DCs type located in the epithelium: the Langerhans cells (LCs). MDA and 3α-OH MDA were topically applied on the ear skin and LCs were quantified on the epidermal sheets after 24 h. Both triterpenes, 3α-OH MDA (65 cells ± 3.5) and AMD (64.8 cells ± 1.6) induced a decrease of epidermal LCs when compared to the untreated control group (73.06 cells ± 0.8, * *p* < 0.0133) (Figure 2). No significant differences were observed between both triterpene-treated groups.

### 2.2. DCs Phenotype after the Treatment with MDA or 3α-OH MDA

The expression of costimulatory and inhibitory molecules defines the extent of DC maturation and the triggering of specific immune responses. To evaluate the MDA and 3α-OH MDA effect on DCs maturation, the expression of the CD40, CD80, and CD86 maturation markers were evaluated. CD273 and CD274 inhibitory markers’ expression was also evaluated (Figure 3 and Figure 4). The treatment with 3α-OH MDA induced an increased maturation and inhibitory markers expression. Regarding the maturation markers, 3α-OH MDA induced a greater CD80 expression (85 ± 5.9) than the without treatment group (WT) (61.22 ± 7) and the MDA-treated cells (72.2 ± 0.3) (*** *p* = 0.0003), it is important to note that the expression of CD80 induced by 3α-OH MDA was similar to that observed in DCs treated with TNFα (84.1 ± 1.2) (Figure 3B). Additionally, 3α-OH MDA caused a higher CD86 expression (77.64 ± 8.9) vs. the WT group (36.5 ± 10.6) (* *p* = 0.04) and similar to TNFα group (88.06 ± 4.5). Moreover, the treatment with 3α-OH MDA increased the expression of the CD274 inhibitory receptor (34.36 ± 3.1) vs. the WT group (17.39 + 5.9) (* *p* = 0.03), although, expression of CD274 in TNFα group (70.6 + 8.7) was higher than DCs/3α-OH MDA (* *p* = 0.03) and MDA (*** *p* < 0.0005). Finally, 3α-OH MDA increased the percentage of DCs expressing the CD86 (82.08% ± 3.3) and CD40 (85.52% ± 2) markers vs. the WT group (CD86: 61.3% ± 3.5; CD40: 65.07% ± 2.6) (** *p* = 0.0052 and *** *p* = 0.0005, respectively), while TNFα produced similar percentages (CD86: 86.3% + 2.6; CD40: 79.6% + 3.8) to 3α-OH MDA and MDA. CD274 MIF for DCs. * *p* = 0.03, *** *p* < 0.0005, **** *p* < 0.0001. These results confirm a 3α-OH MDA-mediated maturation of DCs (Figure 3 and Figure 4).

Conversely, DC changes induced by MDA were different than those induced by 3α-OH MDA. MDA mediated a slight increase in CD80 (30.05 ± 10.2), CD86 (65.8 ± 34.1) and CD40 (47 ± 21) expression compared to the WT group (CD80: 61.22 ± 7; CD86: 36.5 ± 10.6; CD40: 4.3 ± 0.8), although no significative differences were observed. Expression of CD80 (30.05 ± 10.2), CD86 (65.8 ± 34.1) and CD274 (27.56 ± 1.6) were smaller than DCs/3α-OH MDA (CD80: 85 ± 5.9; CD86: 77.64 ± 8.9; CD274: 34.3 ± 3.1) and DCs/ TNFα (CD80: 97.6 + 1.4; CD86: 88.09 ± 4.5; CD274: 70.6 ± 4.3), considered the positive control group (Figure 3 and Figure 4). In addition, a slight increase of CD80+ (70.25% ± 7.8), CD86+ (70.95% ± 6.2), CD40+ DCs percentage (72.8% ± 4.5) in comparison with the WT group (CD80: 56.3% ± 6.2; CD86: 61.30% ± 3.5; CD40: 65.07% ± 2.6). These results suggest that MDA promotes the development of semi-mature DCs (Figure 3 and Figure 4). Besides the phenotype analysis in DCs and in order to evaluate the function of MDA- and 3α-OH MDA-treated DCs, an experimental murine melanoma immunotherapy model was used. Mice with melanoma were inoculated with triterpene-treated DCs, and both tumor growth rate and stroma were subsequently analyzed.

### 2.3. Growth Rate of Tumors from Melanoma Mice Inoculated with Triterpene-Treated DCs

A murine experimental immunotherapy model was used to assess the effect of MDA or 3α-OH MDA on the activity of DCs. Mice with melanoma were treated with DCs previously cultured in the presence of MDA or 3α-OH MDA and tumor growth rate was analyzed afterwards. Several differences were identified after analyzing mice inoculated with MDA- or 3α-OH MDA-treated DCs. Those treated with DCs/3α-OH MDA developed smaller tumors when compared to those developed by mice treated with DCs/MDA (*** *p* < 0.0046) or to the control group (untreated mice) (** *p* < 0.05) (Figure 5D). Conversely, melanoma mice treated with DCs/MDA showed a tumor growth rate very similar to that observed for untreated melanoma mice (Figure 5D). All mice survived twenty-five days after being inoculated with the melanoma cell line.

### 2.4. Histopathological Analysis

Histopathological analysis of the tumor stroma was performed to study the effect of MDA- or 3α-OH MDA-treated DCs. Untreated mice with melanoma showed tumors consisting of basophilic epithelial cells clustered around blood vessels, a particular feature of a high cellular activity. Additionally, a few acidophilic areas comprised by cellular debris and pyknotic nuclei were observed (Figure 5). The tumors identified on melanoma mice inoculated with MDA- or 3α-OH MDA-treated DCs exhibited tumor stroma areas with abundant cellular debris and pyknotic nuclei in a higher proportion when compared to the control group (**** *p* < 0.0001) (Figure 5). No significant differences were observed between the MDA and 3α-OH MDA groups (Figure 5E).

## 3. Discussion

The effect of 3α-OH MDA and MDA on DCs was analyzed in this work. All triterpenes triggered DCs migration, and on the other hand, 3α-OH MDA promoted their maturation, while MDA induced the development of semi-mature DCs. Therefore, when the effect of such triterpenes on DCs function was studied using a murine model of experimental immunotherapy, the group treated with DCs/3α-OH MDA showed a decreased tumor growth rate. In contrast, mice treated with DCs/ MDA showed a tumor growth rate so similar to the control. However, the tumor stroma from mice treated with DCs/3α-OH MDA or DCs/MDA displayed more extensive cell death areas compared to untreated mice.

The effect of triterpenes on the phenotype of DCs was initially analyzed. When DCs were incubated with 3α-OH MDA, they showed high levels of CD80, CD86 and the inhibitory molecule CD274. These are essential molecules for antigen presentation and T lymphocyte activation [2,12]. CD80 and CD86 levels increase when DCs are engaged in antigen recognition and subsequent processing; then, these molecules enhance antigen presentation by DCs to T lymphocytes to trigger an effector immune response, so the maturation of DCs is crucial for the activation and differentiation of naïve T lymphocytes [2]. On the other hand, CD274 is an inhibitory molecule that increases on DCs treated with anti-CD40, LPS, or IFNγ, all identified as molecules that induce DCs maturation, nevertheless, increased expression of CD274 may regulate or even inhibit the activation of T lymphocytes [13]. 3α-OH MDA induced slight increase in the expression of CD274 compared with the DCs without treatment, nonetheless, DCs also displayed high levels of CD86 and CD80, making evident the DCs/3α-OH MDA a mature cells.

Whereas MDA induced their semi-maturation as the levels of costimulatory molecules were higher when compared to untreated DCs, but lower when compared to DCs treated with TNFα, a cytokine that mediates the maturation of DCs. However, no significant changes were detected. When DCs display low levels of both CD80 and CD86 molecules, they are considered immature cells that can trigger a tolerogenic immune response, maintained through a complete lack of stimulatory signals to T lymphocytes during antigen presentation [12,14]. Sometimes, DCs show a semi mature phenotype, characterized by a lower expression of CD80, CD86 and CD40 than matured DCs, with the capacity of migrating to lymph nodes [15]. This phenotype on DCs is comparable to steady-state migratory DCs within the lymphatics, cells that tolerize T cells [2,16].

To our knowledge, this study is the first to analyze the expression of costimulatory molecules in 3α-OH MDA or MDA-treated DCs. In previous investigations, mice were first treated with the *Amphipterygium adstringens* aqueous plant extract and subsequently challenged with dinitrofluorobencene (DNFB). An enhanced T lymphocyte proliferation was observed for mice treated with the extract compared to untreated mice cells [17]. Such aqueous extract contained many molecules, including MDA and 3α-OH MDA. Thus, based on our results, the effect observed on mice with lymphoma was probably caused by the 3α-OH MDA compound occurring in the extract. Additionally, other studies describe an effect of either MDA or 3α-OH MDA on peritoneal macrophages as both compounds triggered nitric oxide (NO) production in resting macrophages, but not in LPS-treated macrophages [10,17]. Macrophages, such as DCs, are antigen-presenting cells (APC), and their NO production after being treated with MDA and 3α-OH MDA suggest that these compounds are involved in APC activation. However, the macrophage costimulatory molecules were not analyzed. In this study, only 3α-OH MDA was able to induce DCs maturation, a momentous event that confers APC with the ability to activate T cells [6,12,15,18]. All of the above is relevant as 3α-OH MDA seems to trigger APC activation and maturation, namely macrophages and DCs.

Moreover, DC migration is another relevant event because APCs migrate to secondary lymphoid organs to induce T-lymphocyte activation after antigen recognition and processing. This study used an in vivo model, where MDA or 3α-OH MDA was topically applied in the mice ear skin to count DCs in the epithelium subsequently. We observed that 3α-OH MDA and MDA stimulated DCs migration. Our data suggest that 3α-OH MDA induced an increased expression of costimulatory molecules, stimulated only one inhibitory molecule, and triggered DCs migration. These results indicate that 3α-OH MDA can induce the DCs activation, thus triggering several antitumor immune responses, including T lymphocytes activation, the primary function of DCs. Furthermore, MDA induced the migration of semi-mature DCs from the skin. As mention above, semi-mature DCs are involved in the differentiation of tolerogenic T lymphocytes that ablate the antitumor immunity [16]. Thus, MDA-treated DCs probably mediate an inhibitory immune response.

In order to demonstrate the effect of triterpenes, an experimental immunotherapy model demonstrated the effect of triterpenes on the DC-mediated immune response. DCs have been proposed as therapy against cancer because they may induce an effector immune response dependent on CD4+ and CD8+ T lymphocytes to eliminate tumor cells [19]. However, to achieve a correct activation of T lymphocytes and the elimination of tumor cells, the DCs must exhibit a mature phenotype, and they should secrete Th1 cytokines. Thus, the effective DCs response may be evaluated by analyzing tumor growth and stroma. It was demonstrated that melanoma mice treated with DCs/3α-OH MDA displayed a lower tumor growth rate when compared to all other groups. This effect implies that from all the triterpenes studied, 3α-OH MDA induced a correct DCs maturation after a 24-h treatment, and it is possible that these DCs activated T lymphocytes and induced the secretion of Th1 cytokines in mice with melanoma. Therefore, as 3α-OH MDA can induce macrophage activation [10], both DC maturation and migration, and the triggering of an immune response able to decrease tumor growth rate, it may be considered a candidate to be used as an adjuvant. It is also possible that 3α-OH MDA may affect the in vivo maturation of resident DCs, macrophages, and lymphocytes.

Moreover, when tumors from mice treated with DCs/3α-OH MDA were microscopically analyzed, they displayed extended tumor cell death areas compared to all other groups. As previously mentioned, DCs/3α-OH MDA may have induced a successful antitumor immune response. Even, it is also possible that the microenvironment produced by DCs and the tumor cell death potentiated the inflammatory response within the tumor stroma, thus inducing the activation of cells such as macrophages and DCs to promote the elimination of different tumor cells and decreasing tumor growth rate [18]. Moreover, previous studies on 3α-OH MDA had similar results when the compound was used to treat prostate cancer xenografts as it decreased the tumor growth rate, promoted the appearance of apoptotic cells, and tumor cell proliferation was also declined [20]. Consequently, 3α-OH MDA may have triggered DCs activation in the Sánchez–Monroy prostate cancer model.

Regarding MDA, the obtained results were not as evident as those observed with 3α-OH MDA. The former did induce a semi-matured phenotype on DCs and, contrastingly, it induced DCs migration. It has been studied that semi-mature DCs are able to migrate to lymph nodes and release immunosuppressive cytokines like IL-10 and TGFβ beta, and stimulate the regulatory T lymphocytes proliferation [21,22]. When the effect of MDA on the activity of DCs to trigger an anti-tumor immune response was studied by using those cells to treat murine melanoma, tumor growth was very similar to that observed in untreated mice. In addition, the tumor cell death areas were more abundant in the group treated with DCs/MDA compared to the group of untreated mice. Similar results were observed when semi-mature DCs were used in the DCs based immunotherapy, such as increased tumor growth rate and diminish survival [23]. Additionally, the presence of semi-matured DCs has been confirmed in cancer patients treated with chemotherapy. In that scenario, in an anti-inflammatory environment, DCs may migrate to the tumor stroma recognizing apoptotic cells resulting in the development of semi-mature DCs able to mediate tumor inhibitory immune responses [15,24]. In summary, all of our results suggest that MDA/DCs mediate DCs semi-maturation, the production of inhibitory cytokines, or a defective antigen presentation [24]. Additionally, the effect of MDA resembles the anti-inflammatory effects induced by the *A. adstringens* extract when tested in a colitis model as it decreased the levels of cytokines such as TNFα and IFNγ and it also restricted tissue damage [11].

It is possible that a higher MDA dose may be needed to induce DCs maturation, although decreased cell viability may be achieved. In this regard, Sanchez-Monroy et al. 2017, used MDA to treat prostate cancer, and their results showed that it was necessary to inoculate a higher MDA dose instead of 3α-OH MDA to observe a positive effect on the tumor growth rate in mice with experimental prostate cancer.

## 4. Materials and Methods

### 4.1. Mice

In all experiments, 6- to 8-week-old C57BL/6 male mice were used. They were kept under light-dark cycles, in a controlled-temperature environment, and they were fed ad libitum within the Animal Facilities of the Department of Cellular and Tissue Biology, School of Medicine, UNAM.

### 4.2. Ethics Statement

The study was approved by the School of Medicine’s Ethics and Research Committees (Comisiones de Investigación y de Ética, dictamen 141/2015), Universidad Nacional Autónoma de México. This study was performed in accordance with the Official Mexican Standard NOM 062-ZOO-1999.

### 4.3. Peptides

The MAGE-AX peptide (LGITYDGM) was synthesized by Research Genetics (Invitrogen, Leiden, The Netherlands) with a 94% purity. The MAGE-AX peptide was stored at −70 °C and it was used at a 25 μg/mL concentration.

### 4.4. Cell Lines

The B16-F10 murine melanoma cell line was used to experimentally induce melanoma in this study (American Type Culture Collection, ATCC, Manasas, VA, USA). These cells were cultured in RPMI-1640 media supplemented with 10% fetal bovine serum (FBS) and 1% penicillin/streptomycin (Gibco). The murine myeloma X63 Ag8.6.5.3 cell line transfected with the GM-CSF gene was used to obtain the granulocyte-macrophage colony stimulating factor (GM-CSF). These cells were kindly donated by Dr. Layla Gutierrez Kobeh (Peripheral Research Unit UNAM-INC, Research Division, School of Medicine, UNAM). The X63 cells were cultured in F12 media supplemented with 10% FBS and 1% penicillin/streptomycin (Gibco).

### 4.5. Antibodies

The monoclonal antibodies used to conduct the flow cytometry analysis are the following: anti-CD11c-allophycocyanin, RRID:AB_313778, clone: N418, cat: 117310, lot: 13206713, Biolegend; anti-Ia/Ie-fluorescein isothiocyanate, RRID:AB_313321, clone: MS/114.15.2, cat: 107606, lot: B199710, Biolegend; anti-CD40-phycoerythrin, RRID:AB_1134084, clone: 3/23, cat: 124610, lot: B197462, Biolegend; anti-CD80-phycoeythrin, RRID:AB_313128, clone: 16-10A1, cat: 104708, Biolegend; anti-CD86-phycoerythrin, RRID:AB_313159, clone: GL-1, cat: 105008, lot: B207219, Biolegend; anti-CD273-phycoerythrin, RRID:AB_2299418, clone: TY25, cat: 107206, lot: B170876, Biolegend; and anti-CD274 phycoerythrin; RRID:AB_2073557, clone: 10F.9 G2, cat: 124308, Biolegend.

### 4.6. Production of a GM-CSF-Enriched Supernatant

A GM-CSF-enriched supernatant was obtained according to Piñón-Zárate et al. 2014 [19], from X63 Ag8.6.5.3 murine myeloma cell cultures transfected with the GM-CSF murine gene. Briefly, cells were cultured in F12 media supplemented with 10% FBS and a 1% antibiotic/antimycotic cocktail. After an 80% confluency, they were harvested by using trypsin/EDTA at 37 °C for five minutes. Afterwards, the cells were re-cultured in F12 media supplemented with the G418 antibiotic, again until an 80% confluency. They were subsequently harvested and seeded on new flasks containing F12 media supplemented with 10% FBS. Finally, when the cells reached an 80% confluency the media was replaced with plain F12 media. The supernatant was collected after 48 h and it was later sterilized by filtration with 0.22 µm filters, frozen, and stored at −70°C until further use.

### 4.7. Generation and Differentiation of Dendritic Cells Obtained from Bone Marrow Cultures

DCs were isolated based on a method previously proposed by Piñón-Zárate et al. 2014 [19]. Briefly, C57BL/6 mice tibiae and femora were collected and they were initially washed by continuous immersion in ethanol and then in sterile Hank’s Balanced Salt Solution (HBSS). Subsequently, bones were perfused with RPMI-1640 media to obtain bone marrow precursor cells that were afterwards incubated in RPMI-1640 media supplemented with FBS, an antibiotic/antimycotic cocktail, and GM-CSF-enriched supernatants of X-63 cells (20%) at 37°C and 5% CO_2_ [19]. Cells were collected after six days and their phenotype was analyzed by flow cytometry.

### 4.8. Cell Migration

To analyze the effect of triterpenes on cells migration, the mice ear skin was treated topically with these compounds. Langerhans cells (LCs) and DCs are located in both the skin and mucosae and they migrate towards the lymph nodes after stimulation. Thus, 0.03 mM MDA or 3α-OH MDA were applied topically to mice on their ear skin. The number of LCs in the epidermal sheets was evaluated after 24 h. A control group was treated with PBS. Briefly, mice were euthanized 24 h after applying the triterpenes and the ear pinnae skin was isolated. LCs were observed at the epidermis by using an indirect immunohistochemical assay to detect CD207 based on a method proposed by Hernández-Segura et al., 2005 [25]. LCs were quantified in micrographs, showing a 1.9 mm^2^ area, from the overall area of 10 ear skin histological sections captured by using a 20× objective on a Nikon Eclipse 80i microscope.

### 4.9. DCs Treatment with MDA or 3α-OH MDA

1 × 10^6^ DCs/mL were incubated in RPMI 1649 media supplemented with 10% FBS and 1% penicillin/streptomycin and they were treated with either 0.03 mM MDA or 3α-OH MDA (diluted in DMSO) for 24 h at 37 °C and 5% CO_2_. The DMSO concentration used for the cultures was 0.1%. DCs incubated with 50 ng/mL of TNFα were considered the positive control group of maturation [19]. Those DCs cultured only in supplemented RPMI media were considered immature.

### 4.10. Assessment of DC Phenotype after the Treatment with MDA or 3α-OH MDA

To evaluate the phenotype of DCs after being treated with MDA or 3α-OH MDA, DCs first washed PBS (0.4 g/L potassium phosphate, 0.726 g/L sodium phosphate and 9 g/L sodium chloride, 0.05% albumin, SIGMA) al 4 °C. Then, cells were stained by the addition of the following antibodies: anti-CD11c allophycocyanin (1:500), anti-Ia/Ie-isothiocyanate (1:250), anti-CD40- phycoerythrin (1:250), anti-CD80- phycoerythrin (1:250), anti-CD86-phycoerythrin (1:300), anti-CD273-phycoerythrin (1:200), and anti-CD274-phycoerythrin (1:400). As negative controls, the following isotype controls were used: rat IgG2b phycoerythrin and rat IgG2a isothiocyanate from Biolegend, San Diego, CA, USA. After the incubation time, all samples were washed with PBS and then fixed it in 0.5% paraformaldehyde in PBS. Overall, 40,000 events were acquired in every assay, based on the expression of CD11c and Ia/Ie on the cells, then, mean intensity fluorescence (MIF) and percentage of DCs positive to costimulatory molecules were analyzed. Data acquisition was performed in a FACScalibur cytometer at the National Laboratory of Flow Cytometry (LabNalCit, Biomedical Research Institute), and the analysis was conducted using the FlowJo software.

### 4.11. Tumor Induction and Tumor Growth Analysis

B16-F10 cells (6 × 10^6^) were subcutaneously inoculated to C57BL/6 mice in order to experimentally induce melanoma. All mice developed a palpable tumor lesion measured by using a Vernier caliper after 15 days. The largest and smallest tumor diameters were measured every other day in order to calculate tumor volume by applying the following equation: V = (A2 × B)/2, where A = smaller diameter and B = larger diameter.

### 4.12. Immunization Protocol

Three experimental groups consisting of ten mice were established to implement the immunization protocol. Fifteen days after inoculating with B16-F10 cells, all mice were treated with DCs (1 × 10^6^.) Experimental groups were divided as follows: (a) untreated, (b) DCs treated with 3α-OH MDA, and (c) DCs treated with MDA. All DCs were also incubated with the MAGE-AX peptide in order to induce a specific antitumor immune response.

### 4.13. H&E Staining of Histological Sections

A histopathological evaluation of all melanomas was conducted according to previously described procedures [26]. All dissected tumors were fixed with a formalin solution (a picric acid saturated solution and 4% buffered formalin) for 24 h. Afterwards, these fixed tumors were embedded in paraffin. Subsequently, up to 10 sections of approximately 5 μm were stained with hematoxylin and eosin for histological examination purposes. The histological study was completed by taking micrographs showing a 1.9 mm^2^ area. The overall area of all 10 histological sections was photographed using a 20× objective on a Nikon Eclipse 80i microscope. The cell death percentage area (acidophilic regions with abundant cells with pyknotic nuclei) was analyzed in each micrograph using the Image Pro Plus 7.0 Media Cybernetics software.

### 4.14. Statistical Analysis

All experiments were performed at least three times by triplicate. Data are shown as the mean value ± SEM. Repeated variance tests (ANOVA) and Tukey post hoc tests were performed in order to evaluate significance after applying the different treatments. *p* < 0.05 was considered statistically significant. All analyses were performed using the GraphPad Prism 6 software.

## 5. Conclusions

In summary, triterpenes studied in the present investigation showed different effects on DCs. On the one hand, 3α-OH MDA promoted the maturation and migration of DCs, essential events to develop an effective anti-tumor immune response, which we could note in the diminish tumor growth rate and tumor stroma analysis in mice with melanoma. On the other hand, DCs incubated with MDA displayed the capacity of migrating and a semi-mature phenotype, related to differentiation and activation of regulatory T lymphocytes and production of inhibitory cytokines. Events that can explain the tumor growth rate observed in mice with melanoma treated with DCs/MDA. As a result, the present research helps to explain the effects of the triterpenes 3α-OH MDA and MDA on DCs and on the immune response against cancer. Taking into account all of the above, only 3α-OH MDA may be considered an immunostimulant or adjuvant based on its effects on DCs, while MDA may be considered a molecule with anti-inflammatory properties.

## Figures and Tables

**Figure 1 molecules-27-01451-f001:**
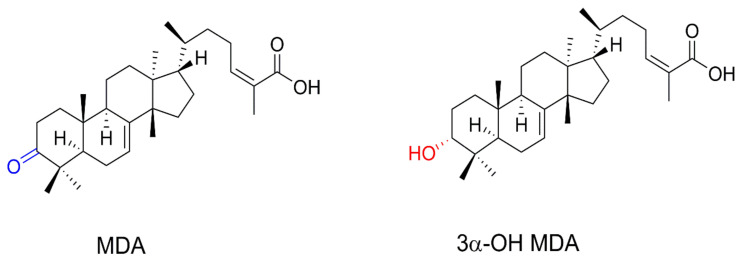
Triterpenes MDA and 3α-OH MDA.

**Figure 2 molecules-27-01451-f002:**
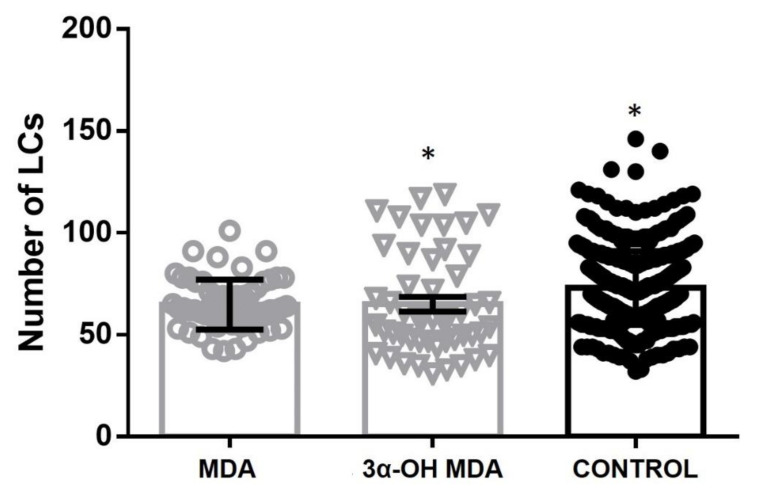
DCs migration after the treatment with MDA or 3α-OH MDA. Treatment with triterpenes decreases the number of DCs in the epidermis compared to untreated mice (control). No statistical differences were observed between MDA and 3α-OH MDA groups. * *p* < 0.01. Mean ± SEM n ≥ 3.

**Figure 3 molecules-27-01451-f003:**
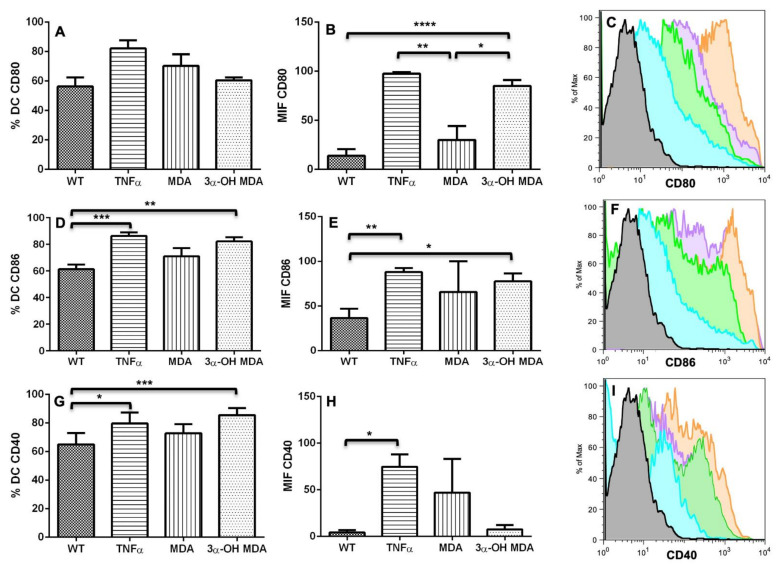
DCs Costimulatory molecules after a treatment with MDA or 3α-OH MDA. The levels of the CD40, CD80, and CD86 costimulatory molecules were measured DCs after a treatment with MDA or 3α-OH MDA. WT: without treatment group and positive control of maturation: TNFα. 3α-OH MDA induces an increased CD80 and CD86 expression and also increased levels of CD40- and CD86-positive DCs. When these cells were treated with MDA, a CD40 increased expression was observed, although no significant differences were observed. (**A**) Percentage of CD80+ DCs after the treatment. (**B**) Mean CD80 fluorescence intensity (MIF). * *p* = 0.03, ** *p =* 0.005, **** *p* < 0.0001. (**C**) Histogram of DCs showing the expression of CD80. Gray: isotype control; Aqua: without treatment (WT: immature DCs); Orange: TNFα; Green: MDA; Purple: 3α-OH MDA. (**D**) Percentage of CD86+ DCs after the treatment. ** *p* = 0.0052, *** *p* = 0.0005. (**E**) CD86 MIF for DCs. * *p* = 0.040, ** *p* = 0.0043. (**F**) Histogram of DCs showing the expression of CD86. Gray: isotype control; Aqua: without treatment (WT: immature DCs); Orange: TNFα; Green: MDA; Purple: 3α-OH MDA. (**G**) Percentage of CD40+ DCs. * *p* = 0.05, *** *p* = 0.0005. (**H**) CD40 MFI for DCs. * *p* = 0.05. Mean ± SEM n > 3. (**I**) Histogram of DCs showing the expression of CD40. Gray: isotype control; Aqua: without treatment (WT: immature DCs); Orange: TNFα; Green: MDA; Purple: 3α-OH MDA.

**Figure 4 molecules-27-01451-f004:**
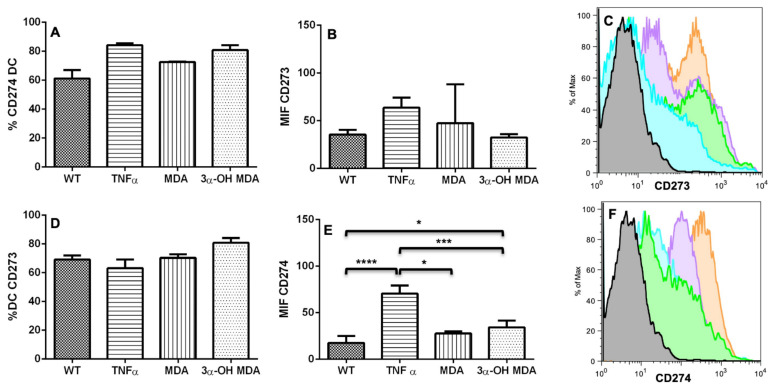
DCs coinhibitory molecules after the treatment with MDA or 3α-OH MDA. The levels of the CD273 and CD274 coinhibitory molecules were measured in DCs after treatment with MDA or 3α-OH MDA. WT: without treatment group and positive control of maturation: TNFα. 3α-OH MDA increased the expression of CD274, although TNFα mediated the highest expression of CD274. (**A**) Percentage of CD273-positive DCs. (**B**) Mean CD273 fluorescence intensity (MIF) for DCs. (**C**) Histogram of DCs showing the expression of CD273. Gray: isotype control; Aqua: without treatment (WT: immature DCs); Orange: TNFα; Green: MDA; Purple: 3α-OH MDA. (**D**) Percentage of CD274-positive DCs. (**E**) CD274 MIF for DCs. * *p* = 0.03, *** *p* < 0.0005, **** *p* < 0.0001. Mean + SEM n ≥ 3. (**F**) Histogram of DCs showing the expression of CD274. Gray: isotype control; Aqua: without treatment (WT: immature DCs); Orange: TNFα; Green: MDA; Purple: 3α-OH MDA.

**Figure 5 molecules-27-01451-f005:**
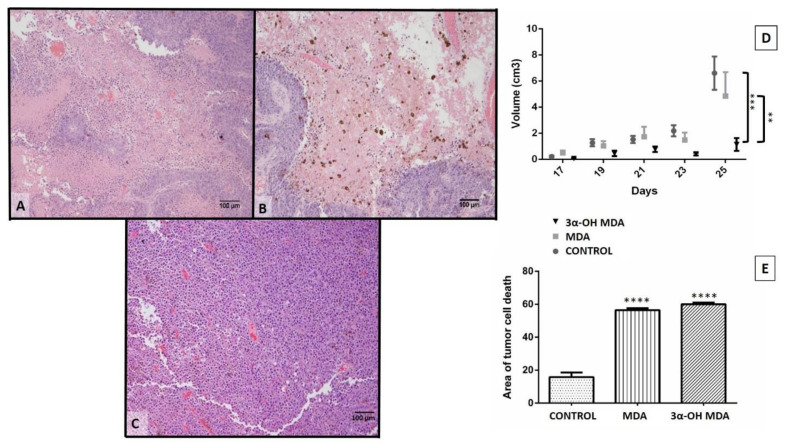
Histopathological analysis and tumor growth rate after a treatment with DCs previously incubated with 3α-OH MDA or MDA. Histological tumors micrographs of mice inoculated with DCs previously treated with MDA (**A**) or 3α-OH MDA (**B**) and untreated mice (**C**). Tumors from untreated mice showed abundant tumor cells and blood vessels. Conversely the MDA and 3α-OH MDA groups showed extended tumor cell death areas consisting of eosinophilic areas (dead cells) and some basophilic areas of active tumor cells. Tumor growth rate changes were measurable up to 25 days after melanoma inoculation. (**D**) Graph showing tumor growth rate in mice treated with DCs previously incubated with MDA or 3α-OH MDA. Mice treated with DCs/3α-OH MD showed the lowest tumor growth rate. Mice inoculated with DCs/MDA showed no differences when compared to the control group. Mean + SEM n ≥ 10. ** *p* < 0.05 Control vs. 3α-OH MDA, *** *p* < 0.0046 MDA vs. 3α-OH MDA. (**E**) Graph showing the change of cell death areas in tumors from mice treated with DCs previously incubated with MDA or 3α-OH MDA. **** *p* < 0.0001, 3α-OH MDA vs. Control, MDA vs. Control. Mean + SEM n ≥ 10.

## Data Availability

The data presented in this study are available on request from the corresponding author.

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
