# Peer review of "Immunomodulatory Properties of Masticadienonic Acid and 3α-Hydroxy Masticadienoic Acid in Dendritic Cells"

_molecules, 2022, doi:10.3390/molecules27041451_

Round 1
Reviewer 1 Report
Dear Authors,
This paper discusses the DCs maturation induced by masticadienonic acid (MDA) or 3α-hydroxy masticadienoic acid (3a-OH MDA). Considering the difficulties that physicians encounter in treating oncological patients any research that provides a possible basis for a better understanding of the mechanisms of tumor growth and secondary dissemination as an unwanted course of the disease is very valuable. Specifically, this work aimed to study the effect of 3α-OH MDA and MDA on DCs used within the context of an experimental immunotherapy model.
The Introduction section clearly what benefit the findings of this study could represent.
The material and methods are precisely and clearly presented. The methodology is rich which contributes to the quality of the results.
Discussion sections discuss the effects of masticadienonic acid (MDA) or 3α-hydroxy masticadienoic acid (3a-OH MDA) on DCs. The results were compared with the available literature data. The conclusion is not so clear, the interpretation of the results is such that it is difficult to conclude whether the test substances have a potential benefit, although findings in this study could be of great clinical importance. Tables and Figures are accurate. References are listed correctly.
I recommend major revision to make the work more understandable to readers because I believe it has scientific potential. For example, the authors stated, “the tumor growth rate induced by DCs treated with MDA showed a tumor growth rate very similar to the control group, so it is proposed that MDA may be a molecule with anti-inflammatory properties”. It is not clear for me why anti-inflamatory when similar to control group. Also, sentences as ” further investigations are needed in order to elucidate additional effects…” are not acceptable.
Best regards
Author Response
Reviewer 1
The reviewer 1 considered: a mayor revision was recommended in order to make more understandable the manuscript to readers, and that the interpretation of the results is such that it is difficult to conclude.
Changes made:
- To clarify the results obtained from the treatment of DCs with 3a-OH MDA and MDA, multiple changes were made in the results section, specifically in section 2.2. In the original manuscript, only three experimental groups could be found: untreated DCs, DCs/MDA, and 3a-OH DCs. Also, a fourth group was added, the positive control of maturation, in which TNFa was used to induce the maturation of DCs. Likewise, representative histograms of each of the costimulatory molecules studied were made. We consider that all the changes help to make the results clearer
On the other hand, in the case of DCs treated with 3a-OH MDA, the data were similar to those found with TNFa, while DCs treated with MDA showed a slight increase in the expression of costimulatory molecules. However, they were never greater than those induced by TNFa. All the changes mentioned above are found in the results section 2.2 and figures 3 and 4 and highlighted yellow.
- Changes were also made to the discussion, mainly trying to explain the effect of MDA on DCs and its possible consequences in more detail. Again, changes were highlighted in yellow.
Revisor 1 commented that: the authors stated, “the tumor growth rate induced by DCs treated with MDA showed a tumor growth rate very similar to the control group, so it is proposed that MDA may be a molecule with anti-inflammatory properties”. It is not clear for me why anti-inflamatory when similar to control group.
Changes made:
- In the original writing, the following sentence could be read: the tumor growth rate induced by DCs treated with MDA showed a tumor growth rate very similar to the control group, so it is proposed that MDA may be a molecule with anti-inflammatory properties.
- Yes, the reviewer has reason: the sentence is unclear. So, to clarify the conclusion, multiple changes were made. First, in the results section 2.2, new data were placed, the related to a fourth experimental group, the positive control, which consisted of the treatment of DCs with TNFα, which induced their maturation. Second, new graphs and histograms were made that allow better analysis of MDA's effect compared with that induced by TNFα and untreated DCs. Finally, in this section it was defined that MDA generated the development of semi-mature DCs. Changes will be found highlighted in yellow.
- Subsequently, in the discussion section, the phenotype of the semi-mature DCs and their effect on the antitumor immune response were explained. It has been confirmed that semi-mature DCs can induce immunological tolerance, promoting tumor growth as in the case of DCs of untreated melanoma mice. Therefore, based on the phenotype observed in DCs treated with MDA and its effect on mice with melanoma, we concluded that MDA could have an anti-inflammatory effect. Changes in the discussion can be seen highlighted in yellow.
Revisor 1 indicated: Also, sentences as ”further investigations are needed in order to elucidate additional effects…” are not acceptable. The conclusion is not so clear.
Changes made:
- The entire conclusion section was changed. An attempt was made to explain better the results obtained and the possible repercussions of the use of triterpenes. According to the data obtained, we suggest the anti-inflammatory effect of MDA and a pro-inflammatory effect by 3α-OH MDA.
Reviewer 2 Report
the aim of the study was to investigate the immunomodulatory action of masticadienonic acid and 3α-hydroxy masticadienoic acid in dendritic cells. The study is nicely conducted; however, there are several extremely important major issues that need to be addressed before publication.
- please provide all details regarding antibodies (dilutions, lot number, RRIDs)
- please provide representative histograms generated from flow cytometry to confirm the data presented on the graphs
- please provide data regarding proper controls in flow cytometry, especially isotype control, the method of staining
- the data from immunostaining needs to be confirmed with western blotting - please provide
- how many samples, how many replicates, how many cells were used for immunostaining
- the discussion would benefit from nice graphics confirming the conclusions
Author Response
Reviewer 2
1. Please provide all details regarding antibodies (dilutions, lot number, RRIDs)
Changes made:
• We appreciate your corrections. We indicated all the details regarding the antibodies adequately in section 4.5 Antibodies. All changes were highlighted in yellow.
- Please provide representative histograms generated from flow cytometry to confirm the data presented on the graphs
Changes made:
- Five representative histograms generated from flow cytometry assays were collocated in figures 3 and 4. As you kindly suggested, the histograms confirmed the phenotypic changes induced by 3α-OH MDA and MDA.
Reviewer 2
- Please provide data regarding proper controls in flow cytometry, especially isotype control, the method of staining.
Changes made:
- Figures 3 and 4 now have five representative histograms generated from flow cytometry from the DCs. Histograms contain data from DCs treated with isotype control, without treatment (immature DCs), TNFα (positive control), MDA, and 3α-OH MDA. Also, figure captions were changed.
- In the results section 2.2, additional data from the positive control group of DCs maturation were added. It will be found all data related to the phenotype of DCs treated with TNFα since it is a cytokine that induces DCs maturation. Also, in section 4.9 was added the concentration of TNFα used in the DCs cultures. All changes are highlighted in yellow.
- The method of DCs staining was completed in section 4.10. Changes are highlighted in yellow.
Reviewer 2
- The data from immunostaining needs to be confirmed with western blotting - please provide
Changes made:
- Thank you for your suggestion. Unfortunately, we do not have western blotting data to confirm the expression of costimulatory molecules in DCs treated with MDA or 3α-OH MDA. However, we believe that the data obtained by flow cytometry allow us to fully understand the effect of both molecules on DCs. On the one hand, flow cytometry allowed us to define a population that was characterized by expressing the Ia/Ie and CD11c molecules; subsequently, it was possible to obtain the mean intensity fluorescence (MIF) of each costimulatory molecule of the Ia/Ie- CD11c population, as well as the percentage of DCs positive to each costimulatory molecule, allowing to know the behavior of a population, in this case, the DCs.
In the case of western blotting, it would not have allowed us to know the exact population that expressed the costimulatory molecules analyzed. Therefore, we consider that the flow cytometric assays shown in the manuscript allow us to study the effect induced by MDA or 3α-OH MDA very well.
Reviewer 2
- How many samples, how many replicates, how many cells were used for immunostaining?
Changes made
- More than three trials were performed with all groups, and in each trial, each group was tested in triplicate. The information was placed in section 4.14, highlighted in yellow.
- In each cell acquisition, 40,000 events were analyzed. The expression of Ia/Ie and CD11c, molecules typical of DCs, was taken into account. Subsequently, the percentage of positive cells for each costimulatory molecule was analyzed, as well as the expression of each one. The information was placed and highlighted in yellow in section 4.10.
Reviewer 2
- The discussion would benefit from nice graphics confirming the conclusions
Changes made
- Figures 3 and 4 were changed. In addition, TNF-induced phenotype data were added, which helped to understand better the effect of MDA and 3α-OH MDA on DCs. It also helped to have placed the representative histograms of each of the co-stimulatory molecules. Consequently, we consider that the results were much clearer.
Round 2
Reviewer 1 Report
I recommend manuscript number Immunomodulatory properties of masticadienonic acid and 3α-hydroxy
masticadienoic acid in dendritic cells for publication in the journal Molecules.
Taking into account the corrections made by the authors in accordance with the recommendations of the reviewers, I suggest manuscript "Immunomodulatory properties of masticadienonic acid and 3α-hydroxy
masticadienoic acid in dendritic cells" for publication without further corrections.
Best regards
Reviewer 2 Report
The Authors improved the manuscript